# Understanding Parenting and Pregnant Women’s Perceptions of Accessing, Utilizing, and Barriers to Seeking Social Support for Mental Well-Being

**DOI:** 10.3390/bs15030348

**Published:** 2025-03-12

**Authors:** Saima Hirani, Monique Sandhu, Nilanga Aki Bandara, Vicky Bungay, Melissa Twomey

**Affiliations:** 1School of Nursing, University of British Columbia, Vancouver, BC V6T 2B5, Canada; vicky.bungay@ubc.ca; 2Faculty of Medicine, Experimental Medicine, University of British Columbia, Vancouver, BC V5Z 1M9, Canada; monique9@student.ubc.ca; 3Department of Medicine, Faculty of Medicine, University of British Columbia, Vancouver, BC V6T 1Z3, Canada; aki101@mail.ubc.ca; 4Mamas for Mamas, Kelowna, BC V1Y 8A6, Canada; melissa@mamasformamas.org

**Keywords:** women’s mental health, mothers, perceived social support, support interventions

## Abstract

Social support has been considered essential for enhancing mental well-being. However, perceptions of support are crucial for promoting psychological well-being, which is shaped by one’s social position across axes of socioeconomic status and gender. This study sought to improve understanding of how pregnant and parenting women living in socioeconomic challenging circumstances perceive social support in order to identify what is working and what needs improvement in mobilizing existing support services. Virtual focus group discussions were conducted with twenty-four women living in British Columbia, Canada. Five themes emerged using thematic analysis: lived realities of motherhood; types of social support women access; barriers they face when accessing support; impact of the pandemic on support services; and women’s recommendations to improve support services. Our findings inform a better understanding of women’s contextual realities in which they live and the need for inclusive supportive approaches for women and their families.

## 1. Introduction

The growing number of psychological issues among women is a serious concern worldwide ([26]). This paper conceptualizes the term women as women who are pregnant, parenting or in caregiving roles. Women’s mental well-being is often amplified through their families because of its intergenerational association with their children’s physical, emotional, and social functioning ([11]; [13]; [24]). Motherhood is a dynamic and demanding time in the lives of many women as they navigate through a multitude of interdependent emotional and physical undertakings. There is a growing recognition that women who are in their reproductive age are more vulnerable to mental health concerns and numerous psychosocial factors can contribute to increased susceptibility to psychological issues ([2]; [27]). Low socioeconomic status and gender are at the forefront of factors correlating with poorer mental health outcomes for women, with food insecurity, poor living conditions and single parenthood presenting as additional complications ([1]). Social support, defined as the mental, emotional, and material resources provided by one’s surrounding social network, is understood to be essential for maintaining mental and physical health ([19]). Evidence also maintains a strong association between social support and improved mental health outcomes, social function and self-efficacy of women, including those living in poor socioeconomic conditions ([6]; [16]). A number of theories have been offered to explain the link between greater social support and better physical and mental health outcomes, with the most prevailing being the stress buffering theory ([7]; [8]). This theory posits social support functions as a buffer, insulating individuals from the adverse impacts of stress on both physical and mental health.

A large body of evidence has demonstrated the imperative of social interventions to promote women’s mental well-being ([3]; [10]; [15]; [25]). However, there exist a range of barriers (economic, cultural, social, and systemic) to these resources, limiting the availability and accessibility of social supports that can foster the wellbeing of both a mother and their children. Further, the two dimensions of social support, namely received and perceived social support, impact the desired outcomes. Received social support (referred to as the quantity or size of supportive behaviours that a person receives) has been found to be less protective against psychological distress as compared to perceived social support, which is defined as the satisfaction with the available support ([9]; [12]; [22]). Therefore, in order to improve women’s mental well-being, it is extremely important to understand their perceptions about the support they receive for themselves and their children and barriers they face while accessing these support sources. Assessing women’s perceived social support also remains critical for developing tailored interventions for promoting women’s and their children’s mental health and overall well-being.

The purpose of this study was to explore parenting and pregnant women’s perceptions of their social support for their mental well-being. We also aimed to unfold factors that women perceive as barriers for seeking support and identify their recommendations for improving support services for their better mental health.

## 2. Materials and Methods

### 2.1. Community Organization Partnership

We partnered with a national charitable organization, Mamas for Mamas https://www.mamasformamas.org/ (accessed on 3 December 2021), based in Canada, which supports mothers and caregivers experiencing poverty-related challenges.

The organization is physically located in Vancouver and Kelowna, British Columbia, and also operates virtually in other parts of Canada with a large online network of women supported through virtual resources. Since 2014, Mamas for Mamas has been striving to support mothers and children by offering them food, clothing and mental health resources and connecting women with available community resources for education, food, housing and mental health services.

They are an inclusive community for mothers and caregivers, which supported over 30,000 Mamas in 2021 directly, with more than 70,000 women partaking in their online sharing economies and satellite branches. The rationale for our emerging partnership is rooted in our shared commitment for improving women’s and their children’s well-being and quality of life. The partnership with Mamas for Mamas facilitated access to a diverse group of mothers (representing various ethnicities, ages, and religions and expecting mothers) and enhanced the credibility and transferability of the findings.

### 2.2. Participants’ Recruitment and Eligibility

Mothers and/or pregnant women who were at least 18 years of age and able to provide informed consent were eligible to participate in this study. Women were recruited through a social media post (which included the research assistant’s contact information) on our community partner’s social media page. Participants who contacted the research team were screened for eligibility and informed about the purpose of the study. Eligible women were invited to attend a virtual focus group discussion. All women participated after signing and submitting a digitalized informed consent form.

### 2.3. Data Collection and Analysis

All participants completed a self-administered socio-demographic online questionnaire using Qualtrics software. Two focus group discussions were held between July and August 2022 and were conducted via the Zoom video conference platform. The sessions were facilitated by *SH* (Principal Investigator) with the help of *MS* (Graduate Research Assistant) lasted approximately 90 min, with 12 participants in each group and a representative from the partner organization. A semi structured interview guide was created in collaboration with the research team and representatives of our partner organization. The interview guide contained a series of questions structured around perceptions and experiences of accessing and seeking social support for mental well-being.

Focus group data were transcribed from the audio recordings of the Zoom meetings. Transcripts were anonymized to ensure confidentiality and audio recordings of the discussion were deleted upon accuracy review of the transcripts. The data were analyzed using the six-step thematic analysis method by [4] ([4]): (i) familiarize with the data; (ii) generate codes; (iii) generate themes; (iv) review themes; (v) define themes; and (vi) locate exemplars.

Based on repeated individual readings of the transcripts by the members of the research team, followed by collective discussion, we created substantive codes updating pre-existing codes as necessary, and identified several emerging themes. The preliminary findings were discussed with the representatives from our partner organization as well to further shape and contextualize the findings in line with socio-ecological approach ([21]). Throughout the analysis process, we followed a continuous, collaborative reflexive approach to acknowledge our biases and understand and interpret the findings through a neutral lens.

### 2.4. Ethical Considerations

The study received ethical approval from the University of British Columbia Behavioural Research Ethics Board. Participants provided informed consent prior to the focus group discussions and were told they could withdraw at any point. To maintain privacy, the participants were given instructions on how to change their Zoom name to a participant ID number in advance of the discussion and were given the option to have their cameras off. To provide additional accommodation, participants were provided an option to type in the chat box if they do not want to respond verbally. Women were also informed that they could refrain from answering questions at any time if they felt uncomfortable.

## 3. Results

A total of 24 women participated in two focus group discussions, with 12 women in each group. This sample size is considered appropriate for the FGDs. The study sample was diverse in terms of age, ethnicity, marital, employment, and education status. Table 1 provides an overview of the demographic characteristics of the participants.

The analysis resulted in five overarching themes: (i) women’s lived realities, (ii) sources of social support, (iii) perceived barriers while accessing or seeking care and support, (iv) impacts of the COVID-19 pandemic on the existing support services, and (v) women’s recommendations for improving support. Figure 1 illustrates a summary of findings including women’s perceptions about the contexts in which they live, sources of social support that they know, access and/or utilize, and barriers that they experience while accessing/utilizing those sources.

### 3.1. Lived Realities of Women

Women described the intricate social contexts in which they were entrenched, shaped by myriad environmental and socio-structural influences. They discussed multifaceted challenges and dilemmas they had to face as they traversed motherhood, having to balance societal expectations with their own mental well-being. One participant expressed the difficulties of no longer taking her psychiatric medications as they were teratogenic and could potentially harm the fetus: “*I want this baby to die because I want to take my medication so badly…I was feeling really guilty*” (P02, pregnant mother). In the postpartum phase, mothers reported feeling unsupported and overlooked, as most of the attention was focused on their child. One mother shared such an experience:
*“I was part of a birthing programme…throughout the pregnancy there was some follow ups, they would check on, like how I’m feeling and if there’s any issues, they would provide resources or give some guidance there. But then beyond that, after the programme ended, there is a lack of support, because you don’t know where to go to find that information”*. (P28, mother)

Mothers reported feeling immense pressure from their inner support circles, their communities, and beyond. As one mother noted, “*Being a mom is super hard. And we kind of I think, I feel a society. We downplay it, how exhausting it can be*” (P10, mother). Furthermore, inadequate or lack of family support was flagged by participants as a significant challenge. One mother shared her struggles, stating, “…*this pregnancy and baby has been really hard. I haven’t had the help. My sisters started working and my mom got sick, so they can’t help with the kids and stuff*” (P11, mother).

Financial constraints can impede access to care for mothers, particularly those who face income disparities and have limited access to resources. As exemplified by one participant who stated, “*I don’t make very much an hour, and I have to essentially work an entire day just to afford an hour in a counselor’s office*” (P10, mother). In addition, some mothers expressed concerns about healthcare professionals lacking sufficient training and resources to provide adequate and validating mental health support. One participant recounted her experience, sharing with the group:
*“I was struggling with thoughts of suicide and self-harm, but I didn’t have access to appropriate resources. Eventually, I realized I was in trouble, and I went to the hospital, only to be told that unless I was actively planning something at that moment, there was nothing they could do”*. (P15, mother)

Some women may be generally unaware of mental health challenges and their support seeking may be hindered by associated stigma. One participant shared, “*I had no knowledge of intrusive thoughts and was unaware of their normalcy. I believed that I had made a significant mistake in my life and questioned my ability to be a mother*” (P15, mother). Women also acknowledged financial challenges and issues associated with being a newcomer in a country as an immigrant or refugee also create complex socioeconomic disadvantages for them and their families.

### 3.2. Sources of Social Support

Social support was identified as a critical aspect that significantly contributes to maternal well-being. We categorized the key sources of support into three groups that women reported they knew, accessed and/or utilized when they needed support: family and friends, community sources, and healthcare services.

Family members, including partners, parents and siblings, are one of the primary sources of support for mothers. A mother’s partner can provide essential emotional and practical support, as exemplified by one mother’s story: “*I broke down to my husband, like two months after having this, his support was honestly one of the biggest things I had*” (P15, mother). Similarly, another participant mentioned her sister who was a stay-at-home mom was a valuable source of support as the participant navigated life as a first-time mom.

Social support from the community was described as another valuable source in ensuring women’s and their children’s well-being. In particular, peer support by other mothers in the community can provide much-needed social support to new mothers, as expressed by one mother: “*being able to talk with other moms who are going through something similar was so reassuring because it’s very much an idea of, okay, we’re not alone*” (P15, mother). Additionally, online groups offer a convenient and accessible space for mothers to connect and receive support, as was the case for many mothers in this study during the COVID-19 pandemic. For instance, one mother found value in taking online parenting classes due to COVID-19 restrictions. Participants also brought up their workplaces that offered them various types of support, such as employee assistance programs, which helped mitigate some of the financial and practical challenges. Women also recognized some tangible support offered by community organizations such as food, clothes, and other necessary goods for their children.

While some participants noted that healthcare providers were not supportive, others shared positive experiences of social support they received from healthcare providers and allied health professionals that made them feel supported and validating, contributing to their overall well-being. One participant shares a positive experience she had with a clinical counsellor, stressing the importance of both availability and accessibility:
*“I suffered from postpartum depression and PTSD. I went to [name of clinic] counselling, and I had a really awesome lady…I’ve never really had a counsellor like that, that was able to let me access her at any time of the day”*. (P09, mother)

Particularly in the absence of other types of social support, healthcare and related care providers can support women by offering empathy and being understanding:
*“I talked to the doctor at [name of maternal health clinic]. And they referred me to a family doctor. And I was able to talk to the doctor… he was really nice and told me, ‘I understand how difficult it can get because, you don’t have family, you don’t have friends, you don’t have anyone.’ And just feeling that someone cared, helped me a lot”*. (P02, pregnant mother)

By providing both emotional support and personalized healthcare, some care providers were identified as making a significant difference in women’s’ lives, helping them navigate through challenging situations and promoting their overall well-being.

### 3.3. Barriers to Accessing and Utilizing Support

Women living in challenging realities face a diverse array of barriers when seeking, accessing, and utilizing social and healthcare support, adding an additional layer of complexity and compounding their existing burdens. Barriers to accessing support were reported at multiple levels, which we categorized as micro-, meso-, and macro-level barriers. By examining barriers through the lens of these levels, we can gain a comprehensive understanding of the intricate factors that impede women’s access to vital systems of social support.

#### 3.3.1. Micro-Level Barriers 

Micro-level barriers are encountered at the point of the individual and their day-to-day experiences within systems and with other individuals. Previous negative experiences with healthcare services were reported as influential factors creating barriers for women. Some may be hesitant to seek support due to prior negative experiences: “*Previously, I had very bad experience seeking help, like for mental illness, which made me kind of like gun shy of trying to reach out again*” (P15, mother). In addition, a few participants shared they were unaware of available resources that could help them access the care they needed, particularly, a participant living in an outskirt area mentioned: “*I didn’t even know that there were online resources available for children’s mental health*” (P12, mother). Women also identified a number of other micro-level barriers, including language barriers for newcomer women, lack of comfort in seeking support, and household financial challenges.

#### 3.3.2. Meso-Level Barriers 

Barriers at the meso-level exist within communities and local institutions. One of the participants emphasized the need for healthcare professionals to take time to empathize with their patients and connect when patients express their concerns: “*Doctors really need to take patients seriously, especially when they’re expressing that they’re struggling… there should be more steps taken to actually find out what it is that is going on with them*” (P26, postnatal mother). Further, the stigma surrounding seeking support was a concern brought forward by some mothers, especially those residing in small, rural communities. One participant noted that in small communities, stigmatization can be a significant barrier to seeking care: “*the community is just too small. And so, fear of judgement is huge. Because everyone knows everybody*” (P10, mother). Other meso-level barriers that women reported included a lack of satisfaction with healthcare and support resources in their respective communities, both urban and rural.

#### 3.3.3. Macro-Level Barriers 

At the structural, economic, and regulatory macro-level, barriers impact the wider social and political context in which women live and go on to further shape meso- and micro-level barriers. Healthcare providers’ shortage was reported as an exacerbating factor for access issues, particularly in remote communities, as one woman noted: “…*there’s just a shortage of people that it just takes her [referring to a rude receptionist] over. If there were more people in the northeast, that will help*” (P21, postnatal mother). Financial constraints at a structural level (e.g., underfunding for chronic mental healthcare and support by the government and workplaces not offering insurance benefits) was noted as a significant barrier to accessing healthcare services for women: “*I didn’t have benefits at the time. So, I couldn’t really afford to just outsource it myself*” (P26, postnatal mother). A few participants also underlined competing pressures on time due to workplace-related duties, which could limit available time for women to seek out necessary care, as one participant noted: “…*you’d have to take a date, like, take time off work to go to the doctor’s and find an appointment and all of that, which was really frustrating and inconvenient*” (P21, postnatal mother]. One woman expressed frustration with depersonalized care she experienced in the landscape of healthcare system: “*It’s kind of like you’re just a number, like get in line…I just went home after basically being rejected…, that’s how it felt because you’re just put on a waitlist*” (P10, mother). Women mentioned frustrations with the lack of available, accessible and consistent healthcare support, including long waitlists and delays in receiving support, inadequate training of healthcare providers, less prioritization of mental health compared to physical health, and a general distrust of the healthcare system as being the key structural barriers in seeking support for their mental well-being.

### 3.4. Dual Impact of the COVID-19 Pandemic on Women: New Resources and Exacerbated Challenges

Our women participants discussed how the COVID-19 pandemic had brought about unprecedented changes in the social supports available to them and their access to healthcare. Some women appreciated the convenience of online services that emerged during the pandemic. One participant noted, “*I do find it a lot easier doing things online where you don’t have to try to pack up your two kids*” (P15, mother). Another woman from a rural community shared, “*I was able to take a lot of parenting classes online. And those classes, they’re actually offered from Vancouver. And it’s only online because of COVID. And I would not be able to take these classes because I live in the north*” (P21, postnatal mother). Furthermore, some women spoke about the convenience of accessing specialist appointments from remote areas, “*I’ve had specialist meetings for my boys because I am like, way up north and stuff’s hard to get to. So, I was able to access doctors and specialists in different towns*” (P1, mother). The development of these new resources has made it easier for some women to access healthcare and educational services for them and their children, which is particularly important for those living in remote areas.

Despite the benefits of new resources, some women experienced several challenges during the pandemic. Some felt that there was a lack of continuity in care and familiarity with service providers as care shifted to virtual platforms: “*I didn’t get to talk to the same doctor. So, I didn’t find that good because I find especially when you’re talking to a therapist, you want to connect with that person and you want to find the right fit*” (P05, mother). Moreover, other women experienced delays in their schedules when accessing healthcare appointments, “*I found that [GPs] weren’t always on time. So, things were delayed, and then you didn’t know what was going on and then you are kind of left in limbo*” (P23, mother). Additionally, a number of our women participants felt that the social interaction provided by online resources was inadequate, as one woman shared: “*it’s online, and then like, you don’t get that social interaction anymore. And some things are just closed*” (P21, postnatal mother). These challenges made it difficult for some women to access the care and support they need during times of crisis.

### 3.5. Women’s Recommendations for Improving Care and Support

Women provided a number of key recommendations in order to improve the existing landscape of care and support interventions. Most of these recommendations centered around various aspects of maternal healthcare, from prenatal checkups to in-home nursing support. They recommended the implementation of enhanced virtual support for first-time mothers, given the steep learning curve they experienced and its impact on their mental health. One mother emphasized the importance of raising awareness about community resources from the beginning: “*one of the first places where this education of resources should start is the doctor’s office when we do prenatal checkup*” (P07, mother).

Another mother suggested that healthcare professionals should be cognizant of the fact that first-time mothers often have limited prior knowledge of caring for their infant and themselves and that support should be provided at home by nurses: “*I think they should be just cognizant, especially for first time mothers who have no clue most of the time, and then provide some support at home by the nurses*” (P07, mother). Women recognized the change in services including mental healthcare as lock down restrictions and policies for virtual care were eased. Some participants suggested continuing hybrid options for these services as they improve access of services especially for people who face difficulties accessing in-person services. Women also highlighted the need for healthcare professionals and community organizations to focus on women’s well-being and be advocates for them. As one participant articulated, “*focus on the mother first to ensure that she’s healing while recovering well, and also, when we strongly say that there’s a pain or there’s something that’s really bothering, I hope they take us seriously*” (P07, mother).

## 4. Discussion

The purpose of this study was to explore parenting and pregnant women’s perceptions of their social support for their mental well-being, the barriers they experience while accessing and/or utilizing social support, and ways in which support services can be improved. Our findings highlighted the complex social challenges that women face (such as poverty, pre-existing mental health issues, burdensome gender expectations, newcomer status in a country, and lack of social support), many of which intersect and contribute to significant hardships. Mothers often prioritize their children’s wellbeing over their own, leading to a range of consequences such as forgoing needed healthcare or facing sleep deprivation due to the demands of motherhood. This phenomenon is consistent with previous research by [14] ([14]), who identified employment, personal goals, and relationships as domains in which mothers make trade-offs to prioritize their children’s needs.

Social support is a critical resource for women to navigate these challenges, and our study found that women receive support from a variety of sources, including family members, healthcare professionals, and community organizations. Women appreciated having access to both in-person and virtual support, similar to the findings of a study conducted by [28] ([28]), which reported that virtual means of support had positive impact on mothers’ and expecting mothers’ perceived level of social support, particularly during the COVID-19 pandemic. Providing women with diverse options for accessing support (online, in-person, hybrid) may enable them to tailor support to their unique needs and preferences. This was most evident for women living in geographically remote areas, i.e., parts of northern British Columbia. Women from these areas may have benefited from the implementation of various virtual resources, as they may not have had access to these resources prior to the pandemic ([5]; [20]).

However, financial and time-related barriers prevent some women from accessing essential resources, such as mental health support. Our study identified the impact of low socioeconomic status on access to mental health resources. These findings are consistent with a study highlighting the overwhelming impact of financial difficulties on single parents’ mental health ([23]). A recent cohort analysis found that mothers’ competing time pressures were significantly associated with financial difficulties, further complicating access to resources for low-income mothers ([18]). These barriers may be aggravated during a stressful time like the pandemic, with the need to balance work and childcare, among a multitude of other responsibilities.

Our study also examined the impact of the COVID-19 pandemic on women’s ability to seek and access support, which had both positive and negative effects. Women appreciated the convenience and accessibility of virtual resources, as noted by [17] ([17]), who found that virtual peer support was helpful for pregnant people and mothers with substance use challenges. However, challenges such as time management and a desire for in-person sessions were also noted, emphasizing the importance of understanding women’s diverse needs and preferences in providing support. Other considerations include barriers to access virtual resources, which may entail not having internet services or a technological device.

Overall, our findings suggest that mothers face complex challenges that require comprehensive and multimodal support interventions to address. Addressing financial and time-related barriers, improving access to mental health resources, and tailoring support to individual needs and desires can help mothers overcome these challenges and promote their health and wellbeing.

### 4.1. Implications for Policy, Practice and Research

A number of recommendations for policy, practice, and research were identified, both by the women and our community partner organization. Women in our study stressed the need for care and service providers and care systems to work with women and not for women, to identify their priorities and understand their needs. This can entail involving women in healthcare and social support service decision-making more and bolstering advocacy in clinical settings. Moreover, it is necessary for women to be engaged as partners in the shared decision-making process.

The systemic issue of inadequate maternal mental health support demands comprehensive policy recommendations. Healthcare professionals, community organizations and government bodies should consider creative, consistent and contextual ways to provide support to women. It is imperative that provincial level initiatives and health authorities advocate and take action to address this issue. One approach is the creation of a centralized resource website that clearly lists all available community resources for diverse geographically located ethnic, age and gender groups, streamlining and collating these resources on a central, user-friendly hub.

It is necessary for this list to be regularly maintained in order to account for new resources or resources that retire. This information should be disseminated to clients and physically handed to them by social workers, nurses, doctors, and any provider of services. Moreover, care providers should undergo training on trauma-informed care, and practitioners who visit mothers should be trained in resource navigation and know when to bring in other service providers or professionals such as social workers for further support and follow-up care.

In-person care should be supplemented by online care, ensuring that women living in remote communities, with travel and financial barriers, and those without access to childcare can still access services. To offer mothers greater access to support, groups for before and after birth should continue to be funded and established, particularly for mothers living in non-urban areas. These groups would help mothers understand what to expect during the various stages of pregnancy and after birth and its associated impact on mental health, and provide a sense of community and comfort for women going through similar experiences. Further work that considers the impact of geography and women’s social support is needed to improve women’s perceptions of support and its positive outcomes on their well-being.

Further, grant restrictions should be reduced to enable agencies to engage in more extensive work and offer long-term services. To promote collaboration and reduce competition among agencies, regular meetings between not-for-profit organizations, government-run facilities, and health authorities should be facilitated. Such meetings would enable stakeholders to remain cognizant of available programs and strategize to fill the gaps to ensure women and families are not left feeling unsupported. Ultimately, these policy recommendations have the potential to substantially improve the women’s mental healthcare experience and outcomes for their children and families.

This study provides key insights into research gaps and suggests that the future research should seek out how new social support programs and interventions could be collaboratively developed to better align with the needs and contexts of women.

### 4.2. Limitations

It is important to note that the perspectives of our women participants may not represent all women including mothers and expecting mothers who experience socioeconomically disadvantaged conditions. However, our participants represent voices of diverse ethnicities, ages, marital, educational and legal residency status. It is also important to acknowledge that our study used an online platform to conduct the FGDs. Although many women appreciated the online medium as it helped them to avoid childcare and commute challenges, it might have hindered the active participation of some women. Our team ensured that our women feel safe and comfortable while participating in the FGDs.

## 5. Conclusions

This study provides insights into women’s experiences and perceptions of social support for their mental well-being and highlights the need for understanding women’s contexts by service and healthcare providers. The study findings inform key sources of support that mothers and pregnant mothers find useful for their own and their children’s mental well-being, and a wide range of barriers they face while accessing and seeking mental health support. These findings also advise the need for inclusive approaches to enhance the awareness, accessibility, and efficacy of these resources to protect and promote mothers’ and their families’ mental health. Further research is needed with a focus on collaboratively designing support interventions that include mothers’ voices and align with mothers and their children’s mental health needs.

## Figures and Tables

**Figure 1 behavsci-15-00348-f001:**
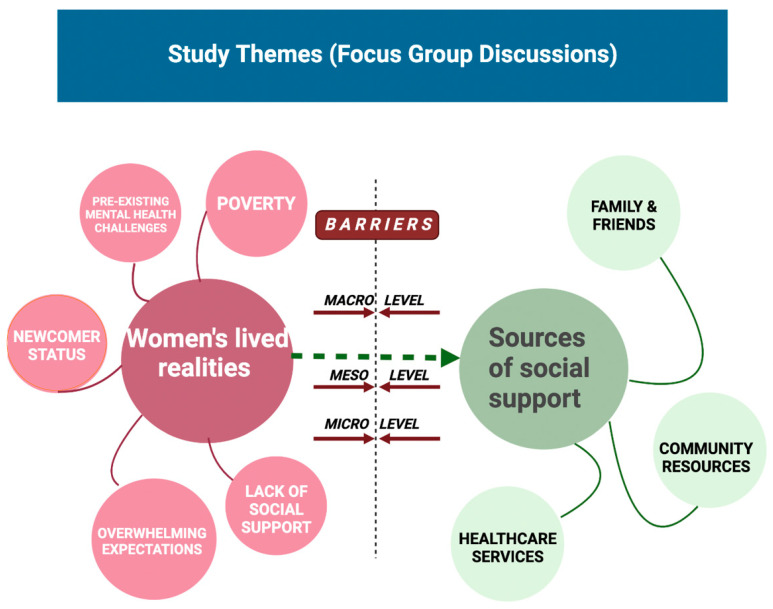
Summary of key themes emerging from focus group discussions.

**Table 1 behavsci-15-00348-t001:** Self-identified demographics of participants (n = 24).

Self-Identified Demographics	Number (Percentage)
Age range	20–49 Years
Ethnicity	
White	16 (67%)
East Asian	3 (12.5%)
South Asian	3 (12.5%)
Latin American	1 (4%)
Indigenous	1 (4%)
Marital Status	
Married	19 (79%)
Single	4 (17%)
Common-law	1 (4%)
Employment Status	
Not working	12 (50%)
Part-time	7 (29%)
Full-time	4 (17%)
Unemployed (prior to COVID-19)	1 (4%)
Highest Level of Education Completed	
Elementary/Grade School	1 (4%)
High School	3 (12.5%)
College/Technical School	12 (50%)
University Undergraduate Degree	5 (21%)
Postgraduate Degree	3 (12.5%)

## Data Availability

The anonymous data supporting the findings of this study are available from the corresponding author upon reasonable request.

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
