# Peer review of "Understanding Parenting and Pregnant Women’s Perceptions of Accessing, Utilizing, and Barriers to Seeking Social Support for Mental Well-Being"

_behavsci, 2025, doi:10.3390/bs15030348_

Round 1

Reviewer 1 Report

Comments and Suggestions for Authors

The article deals with a relevant and sensitive topic to health care, contributing to its better development.
It is well structured, with clear language, well argued and supported by references and current authors.
The results are discussed, and the study's implications for healthcare are defined.
The only aspect I don't understand is that the participants are referred to according to their ethnicity, which is not justified anywhere in the article. I suggest that the participants be referred to in another way to avoid emphasising this variable (ethnicity) without any research purpose. 

Author Response

Comment 1: The article deals with a relevant and sensitive topic to health care, contributing to its better development. It is well structured, with clear language, well argued and supported by references and current authors. The results are discussed, and the study's implications for healthcare are defined.

Response 1: Thank you for your comments.

Comment 2: The only aspect I don't understand is that the participants are referred to according to their ethnicity, which is not justified anywhere in the article. I suggest that the participants be referred to in another way to avoid emphasising this variable (ethnicity) without any research purpose. 

 Response 2:

Thank you for pointing this out. We agree with this comment. We have removed participants’ ethnicity (throughout the quotes) and indicated participants’ pseudo identity numbers with their parenting status, i.e. mother, pregnant mother, and postnatal mother.

Reviewer 2 Report

Comments and Suggestions for Authors

This article addresses the much needed area of information for mothers accessing healthcare. It is very well written and thought out. The only concern that I have is that authors use ethnicity as a reference point (chinese, white, etc). Is there a reason for this instead of using any other marker - age, birth number, income level?

Author Response

Comment 1: This article addresses the much needed area of information for mothers accessing healthcare. It is very well written and thought out. The only concern that I have is that authors use ethnicity as a reference point (chinese, white, etc). Is there a reason for this instead of using any other marker - age, birth number, income level?

Response 1:

Thank you for pointing this out. We agree with this comment. We have removed participants’ ethnicity (throughout the quotes) and indicated participants’ pseudo identity numbers with their parenting status, i.e. mother, pregnant mother, and postnatal mother.

Reviewer 3 Report

Comments and Suggestions for Authors
  • "Myriad" is used incorrectly in places. "...shaped by myriad factors..." NOT "...shaped by a myriad of factors..."
  • I'm not sure why "women" has to be characterized so specifically. Why not just say "pregnant, parenting, and caregiving women"?
  • The research questions or study purpose are not well defined. That shows through later on in the results section because the framework or conceptual model is shaky and seems disorganized, it does not make intuitive sense in the context of the current stated purpose.
  • How does Mamas for Mamas define "inclusive community"?
  • Random ( in line 128
  • In line 147 and then throughout with the quotes, why do the authors have to call out the respondent's ethnicity? Why is that important to the quote when the purpose of the article isn't to make comparisons between ethnic identity groups?
  • Why micro- meso- and macro-level barriers? This should be more clear in the introduction or methods.
  • It seems like the purpose is stated differently at the beginning of the discussion section in a way that changes its meaning.
  • Bringing up the COVID pandemic is tricky because it's not the express purpose of the paper. I wonder if a specific research question about COVID would be useful at the beginning so that it makes more sense in context?
  • Having the focus groups on zoom may have introduced limitations that should be addressed in that section.
  • The authors should consider stating their own biases up front, perhaps through reflexivity.

Author Response

Comment 1: "Myriad" is used incorrectly in places. "...shaped by myriad factors..." NOT "...shaped by a myriad of factors..."

Response 1: Thank you for pointing this out. On page 5, line 142, we correct it to “…shaped by myriad environmental and socio-structural influences.” We also replace myriad on page 2, line 50, with ‘…a range of barriers’.

Comment 2: I'm not sure why "women" has to be characterized so specifically. Why not just say "pregnant, parenting, and caregiving women"?

Response 2: Thank you for this comment. It is a bit unclear if this comment is for any specific section of the manuscript or for the entire paper. In the beginning of the paper (page 2, line 64), we have clearly mentioned that “The purpose of this study was to explore parenting and pregnant women’s perceptions of their social support for their mental well-being.” Therefore, ‘women’ used in the methods and results section actually refer to the participants who were parenting and pregnant women. Parenting and pregnant women/mothers have been used throughout the paper.

However, based on your suggestion and to make it explicit, on page 2, line 86, we have now replaced ‘Mothers or pregnant mothers, who self-identified as women’ with ‘Mothers and/or pregnant women, who were…..’

Comment 3: The research questions or study purpose are not well defined. That shows through later on in the results section because the framework or conceptual model is shaky and seems disorganized, it does not make intuitive sense in the context of the current stated purpose.

Response 3: Thank you for seeking this clarification. On page 2, lines 64-67, we explicitly mentioned that “The purpose of this study was to explore parenting and pregnant women’s perceptions of their social support for their mental well-being. We also aimed to unfold factors that women perceive as barriers to seeking support and identify their recommendations for improving support services for their better mental health.” Figure 1 on page 4 illustrates the overarching themes that emerged from our focus group discussions and addresses the study purpose.

For better clarity, we have now added on page 4, lines 136-139: “Figure 1 illustrates a summary of findings including women’s perceptions about their contexts in which they live, sources of social support that they know, access and /or utilize, and barriers that they experience while accessing/utilizing those sources.”

Comment 4: How does Mamas for Mamas define "inclusive community"? Random (in line 128

Response 4: Mamas for Mamas is an inclusive community for mothers and caregivers offering their services to people regardless of their age, ability, religion and ethnicity.

To make it explicit, on page 2, line 79, we have added ‘They are an inclusive community for mothers and caregivers.’ We have also added Mamas for Mamas’ website link: https://www.mamasformamas.org/  on page 2, line 71.

Comment 5: In line 147 and then throughout with the quotes, why do the authors have to call out the respondent's ethnicity? Why is that important to the quote when the purpose of the article isn't to make comparisons between ethnic identity groups?

Response 5: Thank you for pointing this out. We agree with this comment. We have removed participants’ ethnicity (throughout the quotes) and indicated participants’ pseudo identity numbers with their parenting status, i.e. mother, pregnant mother, and postnatal mother.

Comment 6: Why micro- meso- and macro-level barriers? This should be more clear in the introduction or methods.

Response 6: Our participants reported a wide range of barriers that they experienced at multiple levels while accessing and/or utilizing social support. In order to better understand these barriers, we categorized them as micro- meso- and macro-level barriers.

Comment 7: It seems like the purpose is stated differently at the beginning of the discussion section in a way that changes its meaning.

Response 7: Thank you for bringing this to our attention. We have now rephrased the purpose in the discussion section, to make it aligned with the purpose statement mentioned in the beginning of the paper. In the revised proposal, on page 9, lines 333-336, we added, “The purpose of this study was to explore parenting and pregnant women’s perceptions of their social support for their mental well-being, the barriers they experience while accessing and/or utilizing social support, and ways in which support services can be improved.”

Comment 8: Bringing up the COVID pandemic is tricky because it's not the express purpose of the paper. I wonder if a specific research question about COVID would be useful at the beginning so that it makes more sense in context?

Response 8:  Thank you for seeking this clarification. We agree that there was no specific research question about COVID-19; however, this theme explicitly emerged when we asked women about their perceptions of mental health support. Our research team decided to report this as a separate theme to highlight its significance.

Comment 9: Having the focus groups on zoom may have introduced limitations that should be addressed in that section. The authors should consider stating their own biases up front, perhaps through reflexivity.

Response 9:  Thank you for this suggestion. In the revised proposal, on page 10, lines 429-433, we now added this limitation: “It is also important to acknowledge that our study used an online platform to conduct the FGDs. Although many women appreciated the online medium as it helped them to avoid childcare and commute challenges, it might have hindered the active participation of some women. Our team ensured that our women feel safe and comfortable while participating in the FGDs.”

On page 3, line 117-199, we also mentioned that our team used reflexivity throughout the analysis process. We added, “Throughout the analysis process, we followed a continuous, collaborative reflexive practice to acknowledge our biases and understand and interpret the findings through a neutral lens.”

Round 2

Reviewer 3 Report

Comments and Suggestions for Authors

In my view this manuscript is improved and I support publication.